# Metabolites Concentration in Plasma and Heart Tissue in Relation to High Sensitive Cardiac Troponin T Level in Septic Shock Pigs

**DOI:** 10.3390/metabo12040319

**Published:** 2022-04-02

**Authors:** Bernardo Bollen Pinto, Manuela Ferrario, Antoine Herpain, Laura Brunelli, Karim Bendjelid, Marta Carrara, Roberta Pastorelli

**Affiliations:** 1Department of Acute Medicine, Geneva University Hospitals, 1205 Geneva, Switzerland; bernardo.bollenpinto@hcuge.ch (B.B.P.); karim.bendjelid@hcuge.ch (K.B.); 2Geneva Perioperative Basic, Translational and Clinical Research Group, Geneva University Hospitals, 1205 Geneva, Switzerland; 3Department of Electronics, Information and Bioengineering, Politecnico di Milano, 20133 Milan, Italy; marta.carrara@polimi.it; 4Department of Intensive Care, Erasme University Hospital—Université Libre de Bruxelles, 1070 Brussels, Belgium; antoine.herpain@erasme.ulb.ac.be; 5Experimental Laboratory of Intensive Care—Erasme University Hospital, Université Libre de Bruxelles, 1070 Brussels, Belgium; 6Laboratory of Mass Spectrometry, Department of Environmental Health Sciences, Istituto di Ricerche Farmacologiche Mario Negri IRCCS, 20156 Milan, Italy; laura.brunelli@marionegri.it (L.B.); roberta.pastorelli@marionegri.it (R.P.); 7Department of Anesthesiology, Pharmacology and Intensive Care, Geneva Hemodynamic Research Group, Geneva University Hospitals, 1205 Geneva, Switzerland

**Keywords:** cardiac dysfunction, myocardial injury, cardiac troponin T, sepsis, septic cardiomyopathy, hydroxyproline, proline, collagen

## Abstract

Elevated circulating cardiac troponin T (cTnT) is frequent in septic shock patients. Signs of myocardial ischemia and myocyte necrosis are not universally present, but the precise mechanism for elevated cTnT is unknown. We investigated plasma and heart tissue metabolites concentration in six septic shock (SS) and three sham swine undergoing a protocol of polymicrobial septic shock and resuscitation, in order to highlight possible pathways and biomarkers involved in troponin release (high sensitive cardiac troponin T, hs-cTnT). The animals were divided into two groups: the high cTnT group (n = 3) were pigs showing a significantly higher concentration of cTnT and lactate after resuscitation; the low cTnT group (n = 6, three sham and three septic shock) characterized by a lower value of cTnT and a lactate level < 2 mmol/L. Spearman correlation was assessed on plasma fold-change of cTnT, cytokines (TNF-α and IL-10), and metabolites. Finally, the fold-change between the end of resuscitation and baseline values (Res./BL) of plasma metabolites was used to perform a partial least square discriminant analysis (PLS-DA) with three latent variables. Before building the model, the number of features was reduced by summing up the metabolites of the same class that resulted similarly correlated to cTnT fold-change. Proline and glycine were significantly higher in the high cTnT group at the end of experiment both in the myocardium and plasma analyses. Moreover, plasma proline fold-change was found to be positively correlated with cTnT and cytokine fold-changes, and trans-4-hydroxyproline (t4-OH-Pro) fold-change was positively correlated with cTnT fold-change. The PLS-DA model was able to separate the two groups and, among the first ranked features based on VIP score, we found sugars, t4-OH-Pro, proline, creatinine, total amount of sphingomyelins, and glycine. Proline, t4-OH-Pro, and glycine are very abundant in collagen, and our results may suggest that collagen degradation could represent a possible mechanism contributing to septic myocardial injury. The common phenotype of septic cardiomyopathy could be associated to dysregulated collagen metabolism and/or degradation, further exacerbated by higher inflammation and oxidative stress.

## 1. Introduction

Sepsis can be defined as a dysregulated response to infection. Approximately one third to one half of patients with sepsis or septic shock develop an impairment of cardiac performance [1,2].

In vitro studies showed that cytokines have a cardio-depressant effect [3,4]. However, the translation of these phenomena into in vivo clinical studies has not been demonstrated [5] and the full elucidation of the mechanisms responsible for septic myocardial dysfunction is lacking [6].

Cardiac troponins are intra-cellular regulatory proteins that control the calcium-mediated interaction of actin and myosin. In particular, cardiac troponin T (cTnT) binds to tropomyosin, facilitating contraction, and is abundant in the heart tissue. The release of cardiac troponins in the circulation can occur when myocytes are damaged by a variety of conditions, such as ischemia, trauma, exposure to toxins, inflammation, and strenuous exercise [7,8].

Sepsis is frequently associated with release of cardiac troponins and high levels of circulating cardiac troponins are independently associated with in-hospital and short-term mortality after sepsis [9,10,11]. In the majority of septic patients, cardiac troponin elevation occurs in the absence of acute coronary syndromes and clinically detectable myocyte necrosis. Moreover, myocardial depression is a reversible process in most surviving patients [12,13]. Left ventricular diastolic dysfunction and right ventricular dilation were found to be the echocardiographic variables correlating best with concomitant high-sensitivity cardiac troponin T (hs-cTnT) concentrations [14]. In addition, structural signs of myocardial necrosis are absent in multiple pre-clinical models of sepsis [15]. Elevated cTnT could be a sign of reversible damage to the contractile complex of heart muscle cells due to a transient insult or a more subtle alteration of cardiac tissue properties, which can lead to cardiac dysfunction later on [16]. Similarly to post endurance exercise, elevated cTnT could be a sign of cardiac stress [17]. After exercise, the circulating level of troponin rises, often returning to the normal range within 24 h, and this is not associated with worse long-term prognosis, suggesting that myocardial necrosis and cell death do not play a major role.

In previous works [18,19], we studied a polymicrobial peritonitis porcine model with an experimental protocol that was originally designed only to investigate the metabolic and hemodynamic changes induced by septic shock, and not with the objective to induce acute heart failure or myocardial dysfunction. However, we noticed that, at the end of the experiment, after full resuscitation, the values of circulating hs-cTnT were elevated in a subset of septic animals and higher than those reported in sham animals.

This observation questioned us about the fact that myocardial stress may occur in a very early phase of sepsis before any clinical manifestation of cardiac dysfunction. For this reason, we decided to further investigate the plasma metabolite concentrations in order to shed light on possible pathways involved in the release of cardiac troponins in this subgroup of septic animals in comparison with the sham and the other septic pigs. We searched for possible associations between the elevation of circulating hs-cTnT and the changes in plasma metabolites and metabolomic profile of the heart tissues provided by quantitative mass spectrometry-based target metabolomics. This is an ancillary study, and the previous hemodynamic and plasma metabolomics data are integrated here with the metabolomic analyses of the harvested heart tissues.

## 2. Methods

### 2.1. Animal Study

The animal study was performed according to the EU Directive 2010/63/EU for animal experiments and the ARRIVE guidelines for animal research and was approved by the local animal ethics committee (Comité Ethique du Bien Être Animal, protocol 641 N). Details on the animals, anesthesia, instrumentation, and experimental protocol were previously reported [18,19].

Nine adult pigs (Sus scrofa domesticus) of both sexes, aged 4–6 months, with a mean (±SD) weight of 42.3 ± 3.9 kg, were instrumented in the Experimental Laboratory of Intensive Care (LA1230336) at the Université Libre de Bruxelles.

The animals were sedated in their cage with an intramuscular injection of 1.5 mg/kg midazolam (Dormicum; Roche, Belgium) and 5 mg/kg azaperone (Stresnil, Eli Lilly Benelux, Brussels, Belgium). The animals underwent endotracheal intubation following the induction of anesthesia with an intravenous injection of 3 mg/kg sufentanil (Sufenta Forte, Janssen-Cilag, Beerse, Belgium), 1 mg/kg propofol (Propovet, Zoetis, Ottignies-Louvain-la-Neuve, Belgium), and 0.5 mg/kg of rocuronium (Esmeron, Organon, Oss, The Netherlands).

Mechanical ventilation was performed in volume-controlled mode (Primus, Dräger, Lübeck, Germany) with tidal volumes of 8 mL/kg and a PEEP set at 5 cm H_2_O.

Fluid maintenance was provided by a balanced crystalloid (Plasmalyte, Baxter, Lessines, Belgium) perfusion at 3–5 mL/kg/h, aiming for a normal volemia at baseline, according to an arterial pulse pressure variation (PPV) < 12% (and a negative fluid challenge in case of doubts, i.e., PPV remaining in the gray zone). A 7 Fr introducer was inserted into the right external jugular vein and a pulmonary artery catheter (CCO; Edwards LifeSciences, Irvine, CA, USA) was advanced in a pulmonary artery for continuous cardiac output (CO), right heart pressures, and mixed venous oxygen saturation monitoring. Both femoral artery pressure and pulmonary artery pressure signals were continuously displayed (SC9000, Siemens, Munich, Germany) and exported to an A/D recording station (Notocord Hem, Notocord, Le Pecq, France), similarly to ECG and cardiac output signals.

Two hours after the end of instrumentation, baseline measurements and blood samples were obtained (baseline, BL). Animals were then randomized to sepsis induction with an intraperitoneal instillation of 3 g/kg of autologous feces and fluid maintenance reduction (1 mL/kg/h) (septic shock group, SS, n = 6); or to a sham procedure (sham group, n = 3). After sepsis onset in the SS group and approximately 1 h with a radial mean arterial pressure (MAP) between 45 and 50 mmHg, a second set of measurements was performed (septic shock, SS). Next, fluid resuscitation was resumed, with both balanced crystalloids and colloids (Geloplasma, Fresenius Kabi, Cabourg, France), still aiming at a PPV < 12%. After 2 h of hemodynamic stability (defined by a stable MAP with no further increase in cardiac output to a fluid challenge), an infusion of norepinephrine at 0.3 µg/kg/min (Noradrenaline tartrate, Aguettant, Lyon, Belgium) was administered, and after 1 h a set of measurements was performed (full resuscitation, Res.). The same time points were defined in the sham group taking in account our historical average timing of sepsis onset and septic shock development in this model (roughly 6 h). All animals were then euthanized with a lethal injection of potassium chloride and an overdose of thiopental. Thereafter, heart samples (right atrium RA, right ventricle RV and left ventricle LV) were rapidly collected according to the protocol for metabolomics analysis. At each time point, arterial blood samples were collected, and EDTA-plasma isolated for metabolomics and laboratory analyses, after immediate refrigerated centrifugation. Samples were kept at −80 °C until analysis.

### 2.2. Plasma and Heart Tissue Analyses

A targeted quantitative approach using a combined direct flow injection and liquid chromatography (LC) tandem mass spectrometry (MS/MS) assay (AbsoluteIDQ 180 kit, Biocrates, Innsbruck, Austria) was applied for the metabolomics analysis of plasma and heart tissues from swine models. The assay quantifies 186 metabolites from six analyte groups: acylcarnitines, amino acids, biogenic amines, hexoses (sum of hexoses), glycerophospholipids and sphingomyelins. The metabolite extracts were processed following the instructions by the manufacturer and analyzed on a triple-quadropole mass spectrometer (AB SCIEX triple-quad 5500, SCIEX Headquarters, Framingham, MA 01701, USA) operating in the multiple reaction monitoring (MRM-MS) mode. Methodological details and data pre-processing have been extensively reported in our previous articles [18,20]. Measurable metabolites and their extended names and abbreviations are listed in Appendix A. The levels of all the measured metabolites (µM) are reported in Appendix A.

Hs-cTnT was measured using an Elecsys assay from Roche Diagnostics. The plasma concentration of cytokines (TNF-α and IL-10) was measured by the specific ELISA kits from R&D Systems.

### 2.3. Data Analyses and Statistics

We stratified the animals by considering the hs-cTnT concentration at the end of the experiment. Hence, 3 SS pigs out of 6 had the highest values of hs-cTnT, higher than sham pig, and a lactate level greater than 2 mmol/L (high hs-cTnT group), whereas the other 3 SS pigs and 3 sham pigs had lower values both of hs-cTnT and lactate (low hs-cTnT group). See Figure 1.

We compared the metabolite concentrations in the heart tissues between the two groups (high vs. low cTnT group) by using Wilcoxon rank-sum test. Successively, we assessed the fold changes in plasma metabolite concentration as the ratio between the values assessed after shock development and baseline (SS/BL) and between the values after full resuscitation and baseline (Res./BL). We then analyzed the correlation between the fold change of metabolite plasma concentrations and the fold change of hscTnT and cytokines by means of the Spearman correlation (all animals n = 9). Significance was considered with a *p*-value < 0.05. These correlations are meant to describe/illustrate the changes in the metabolite concentrations in association with the changes in hs-cTnT induced by the inflammatory insult.

Finally, we used the fold change Res./BL of the plasma metabolite concentrations to perform a partial least square discriminant analysis (PLS-DA) with 3 latent variables. In particular, we reduced the number of metabolites to 36 features by summing the metabolites of the same class that resulted similarly correlated to the fold change in hs-cTnT. The features consisted in the sum of lysophosphatidylcholine 16 and 18 (lysoPC16 + 18), the sum of all measured sphingomyelins species (SM), the sum of all measured phosphatidylcholines species (PC), the individual amino acids, biogenic amines, and sugars.

## 3. Results

A complete description of the hemodynamic and laboratory parameters, including cytokines and hs-cTnT, for all six animals in the septic shock (SS) group were previously reported [18,19]. The criteria adopted for the classification of the animals are illustrated in Figure 1. Three SS were assigned to the high cTnT group (hs-cTnT > 35 ng/L and lactate > 2 mmol/L) whereas the other three SS pigs and three sham pigs to the low cTnT group.

The hemodynamic and laboratory data are illustrated in Table 1 for the two groups (high versus low cTnT). In Appendix A, we report the same parameters for the septic animals only. The three pigs in the high hs-cTnT group had higher HR and CO after full development of shock, and lower O_2_ saturation, PaO_2_, and blood pressure at the end of the experiment with respect the other septic animals (Appendix A). No clinical sign of myocardial infarction (including echocardiography wall motion abnormalities or contractility reduction) was detected in any of the septic pigs.

We compared the metabolites concentration in heart tissues between the two groups, i.e., high vs. low hs-cTnT (Figure 2, Appendix A). There was a significant difference in left ventricle, right ventricle, and right atria for some PC species. Interestingly, proline was found to be significantly higher in the high cTnT group in all heart tissues, whereas glycine was significantly higher in RA and RV tissues.

We investigated the correlation between the fold-change in hs-cTnT concentration in peripheral blood with the fold-change in both cytokines and metabolite concentrations in all animals (n = 9). Figure 3 and Appendix A illustrate the significant correlations for Res./BL and SS/BL fold-change, respectively. Most of the fold-changes between the metabolites level measured at baseline and after full resuscitation (Res./BL) are correlated with the fold-change in hs-cTnT (Figure 3). In particular, the fold-change of hs-cTnT was positively correlated with cytokines, creatinine, trans-4-hydroxyproline (t4-OH-Pro), and proline (Pro) and negatively correlated with sphingomyelins (SM), the most abundant lysophosphatidylcholine species (lyso PC 16 and lyso PC 18), and most diacylphosphatidylcholine species (PC). In addition, the fold-change of both cytokines was positively correlated with proline fold-change. The high level of correlation observed among the species of SM, PC, and lysoPCs and hs-cTnT values suggests they likely carry similar information.

Figure 4 shows the classification obtained from the PLS-DA. The model produces a perfect separation between the two groups. The VIP scores of the model are reported in Appendix A, where the first ranked features are: sugars, hydroxyproline (t4-OH-Pro), proline (Pro), creatinine, total amount of sphingomyelins (SM), glycine (Gly).

Moreover, Figure 4 shows the distribution of values of those metabolites that resulted significantly different between the two groups at the end of the experiment after full resuscitation. Among these features, we can notice again hydroxyproline (t4-OH-Pro), proline (Pro), and creatinine.

## 4. Discussion

In a swine model of resuscitated hyperdynamic septic shock, we identified a subset of septic animals with elevated circulating hs-cTnT and lactate levels (>2 mmol/L). None of the septic pigs had clinical evidence of heart failure or altered cardiac performance evaluated from common echo inspection performed during the experiment. Although the experiment was not designed to induce heart failure and the septic shock condition was reached in all the pigs in SS group (i.e., radial MAP between 45–50 mmHg), in the high hs-cTnT group, the same insult induced a particular phenotype with higher HR and CO and more prolonged hypoxia and hypotension (lower value of oxygen saturation, PaO_2_, and blood pressure at the end of the experiment) (Appendix A).

The metabolite concentrations in different heart tissues were compared between the two groups and proline concentration was found to be significantly higher in all heart tissues and glycine concentration to be significantly higher in right ventricle and right atria in the septic animals with the highest values of hs-cTnT.

A multivariate model was built on the fold-changes (Res./BL) of plasma metabolites levels, and this was able to separate the two groups. Among the first ranked features we have: sugars, trans-4-hydroxyproline (t4-OH-Pro), proline (Pro), creatinine, the total amount of SM species, and glycine.

By considering the fold changes in plasma metabolite concentrations, proline was found to be positively correlated with hs-cTnT and cytokine fold-change, whereas trans-4-hydroxyproline and creatinine were found to be positively correlated with hs-cTnT fold-change only.

In the panel of metabolites that resulted in the first ranking positions according to VIP scores of the multivariate model, we observed that proline and trans-4-hydroxyproline were significantly higher after the full resuscitation in the high hs-cTnT group.

Proline, hydroxyproline, and glycine are abundant in collagen [21]. The chemical structure of collagen is rich in glycine, proline, and hydroxyproline in the repeated form of tripeptides (glycine–proline-Y and glycine–X- hydroxyproline, where X and Y are any amino acid). Moreover, the relative abundance of hydroxyproline in collagen, compared with other proteins, makes hydroxyproline a specific marker of collagen content and turnover.

The results obtained from the analyses of metabolite concentration on plasma and heart tissue suggest a possible disturbed collagen homeostasis/turnover and/or collagen release in sepsis. Collagen is one of the most abundant proteins in the cardiac extracellular matrix [22]. The complex collagen three-dimensional weave makes individual myocytes interconnected through a collagen–integrin–cytoskeletal–myofibril network. This network acts as mechanical support for cardiac myocytes during contraction and relaxation and also helps in the mechanical transmission of individual myocyte shortening into ventricular contraction [23]. The cytoskeleton forms the scaffold of cardiomyocytes as it regulates cell shape, provides mechanical integrity and resistance, and stabilizes the sarcomeric proteins. Moreover, the structural framework mediates biomechanical and biochemical signalling. This provides a plausible mechanism for the elevation of cardiac troponin in the bloodstream in the absence of irreversible cardiac myocyte necrosis.

The mechanisms underlying cTnT release not associated with myocardial ischemia need to be further elucidated. For example, after endurance exercise, cardiac cTnT elevations usually return to within the normal range within 24 h, but some endurance athletes can develop cardiac comorbidities in the long term, such as cardiac structural remodelling with signs of fibrosis of the interventricular septum [24].

In heart failure patients, an association between myocardial injury measured by hs-cTnT and biomarkers of collagen synthesis and degradation was observed in a longitudinal study [25]. The authors conclude that myocardial injury is associated with pathophysiological processes that are critically involved in tissue repair and fibrosis. Such findings may be related to collagen degradation and/or a decrease in collagen synthesis. The authors highlighted that myocardial edema and collagen loss affect myocardial stiffness in opposing manners (edema increases stiffness, whereas collagen loss decreases stiffness).

Collagen metabolism has been associated to inflammation and oxidative stress in several studies. For example, in [26,27], the authors tested the hypothesis that inflammatory cytokines exert a net negative effect on collagen turnover by cardiac fibroblasts. In those in vitro studies, IL-1β and TNF-α each decreased collagen synthesis and increased matrix metalloproteinases (MMPs) activity. Decreased collagen synthesis was associated with the increased expression of the mRNAs for proMMP-2 and proMMP-3. In addition, the authors demonstrated that oxidative stress activated MMPs and decreased fibrillar collagen synthesis in cardiac fibroblasts in vivo. In a rodent model of sepsis, Yu P et al. [28] found that the percentage volume fraction of interstitial space in heart tissue (i.e., myocardial edema) was significantly increased and myocardial collagen content, as assessed by conventional histology quantification, was decreased in septic animals in comparison with sham at 24 and 48 h after the insult.

Collagen degradation could be an important mechanism that contributes to myocardial injury in patients with septic shock. In a longitudinal study of Gaddnas et al. [29], markers of collagen metabolism were increased in patients with sepsis. The authors showed that increasing serum collagen propeptide levels were associated with the development of multiple organ failure and death in patients with sepsis, whereas in survivors all the values had returned to the normal range and were lower at three and six months compared to those at the beginning of the study. The authors inferred that fibrosis may be a central mechanism in the pathogenesis of multiple organ dysfunction. However, the analyses were performed from serum only, as heart tissue was not available, and this did not help in explaining the pathophysiological mechanisms at the myocardial level. In another study, patients with SS were followed for 28 days and biochemical samples were collected up to six days after sepsis diagnosis. Patients with high circulating cardiac troponin were found to be at higher risk of death. The heart of patients who died were harvested and the authors performed histology and immunohistochemistry of the myocardial tissue, information not commonly available [30]. In non-survivors, there was significant myocardial damage, inflammatory cell infiltration, increased collagen deposition, and derangement of mitochondrial cristae.

Given that our animal experiment had an observational time window after shock onset of about 3 h, we hypothesized that possible metabolic derangements occurring in the very early phase of septic shock could be one of the concurrent causes to sepsis induced myocardial dysfunction or cardiac remodelling, which are often observed in septic patients.

Further research is needed to fully characterize the association between collagen turnover and cardiac dysfunction or remodeling in sepsis. Clinical studies with serial measurements of markers of collagen synthesis and degradation and advanced imaging techniques to quantify cardiac dysfunction, fibrosis, and/or edema (e.g., cardiac MRI, speckle-tracking echocardiography) could establish the temporal relation between the two phenomena. More detailed evidence could be provided by preclinical studies of sepsis-induced heart failure with serial cardiac biopsies. However, in order to establish a causal link, the modulation of cardiac collagen metabolism should result in significant changes in circulating cTnT and cardiac function.

We are aware of several limitations of this work. First of all, we can list the small number of animals used, inherent to the ethical and logistic constraints in the pre-clinical research on large animals. Secondly, the original experimental swine study was designed to study the metabolic and hemodynamic changes induced by the early phase of circulatory shock and not to induce per se a condition of functional impairment with an acute heart failure within the time frame of the experimental observations. Third, the increase in creatinine in the high hs-cTnT group could suggest compromised kidney function and the high level of cardiac troponin could be affected by reduced kidney clearance [31]. Finally, although proline concentration was found to be significantly higher in all heart tissues in the high hs-cTnT group, the information related to changes in metabolite concentrations in plasma cannot prove a direct association between collagen mechanisms and the change in hs-cTn. Indeed, plasma proline and hydroxyproline abundance may not necessarily be related to heart-derived collagen degradation, as collagen is a constituent of vascular walls and other organs tissues, for example.

## 5. Conclusions

The main finding of this work is that, in a swine model of hyperdynamic septic shock without overt signs of heart failure, elevated values of circulating cTnT were present within a couple of hours of shock onset in a subset of septic pigs and the hs-cTnT level was found to be associated with metabolites that may be related to collagen metabolism. This may suggest that myocardial injury and septic cardiomyopathy could be the result of dysregulated myocardial collagen metabolism and/or collagen degradation starting in the early phase of shock. Furthermore, our observations indicate that mechanisms leading to secondary cardiac remodelling could start concurrently with the well-known symptoms of septic shock. Further studies are needed to confirm this hypothesis.

## Figures and Tables

**Figure 1 metabolites-12-00319-f001:**
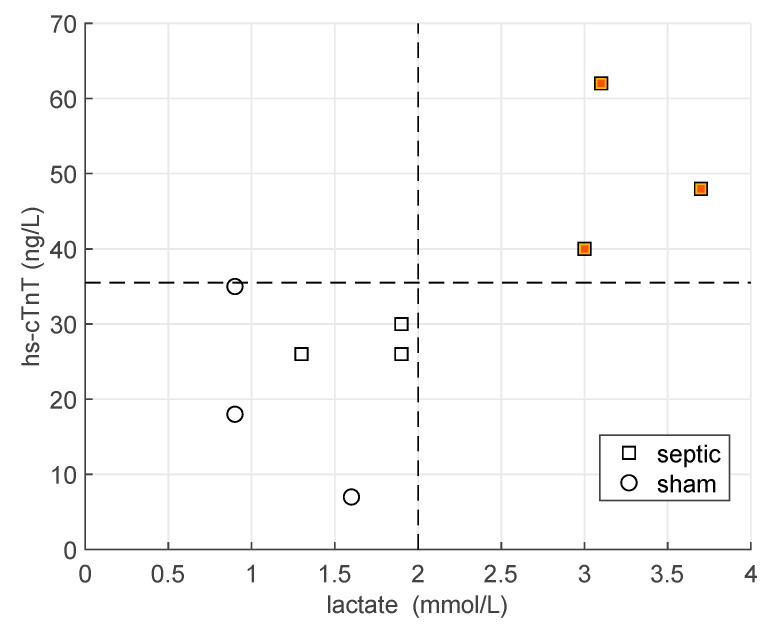
Values of hs-cTnT and lactate at the end of the experiment for all the animals (n = 9). Squares refer to septic pigs, whereas circles to sham animals. The black lines mark the levels of hs-cTnT = 35 ng/L and lactate = 2 mmol/L. The orange squares represent the high hs-cTnT group.

**Figure 2 metabolites-12-00319-f002:**
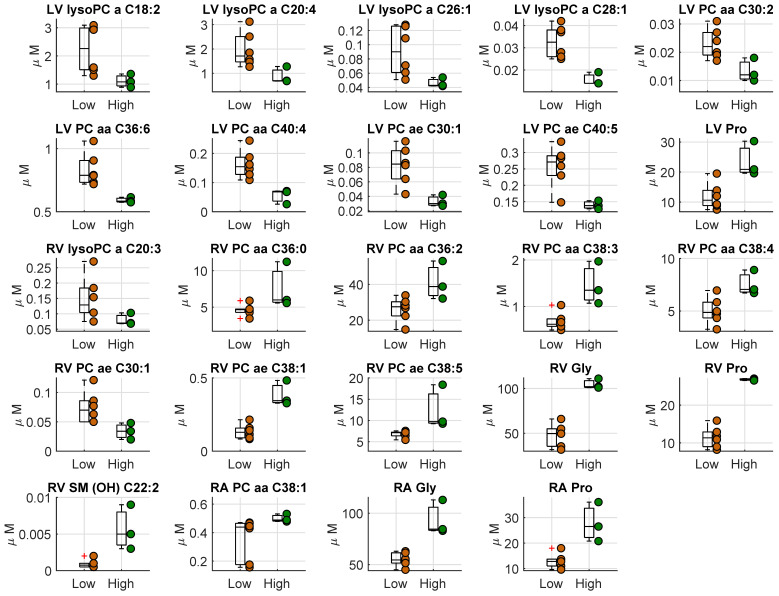
Boxplot of the metabolites concentration (µM) which resulted significantly different in the three different heart tissues (LV, left ventricle; RV, right ventricle; RA, right atrium). The two groups consist in the septic animals with a high hs-cTnT and lactate > 2 mmol at the end of the experiment (high hs-cTnT, reported in green) and the other septic and sham animals (low hs-cTnT, reported in orange).

**Figure 3 metabolites-12-00319-f003:**
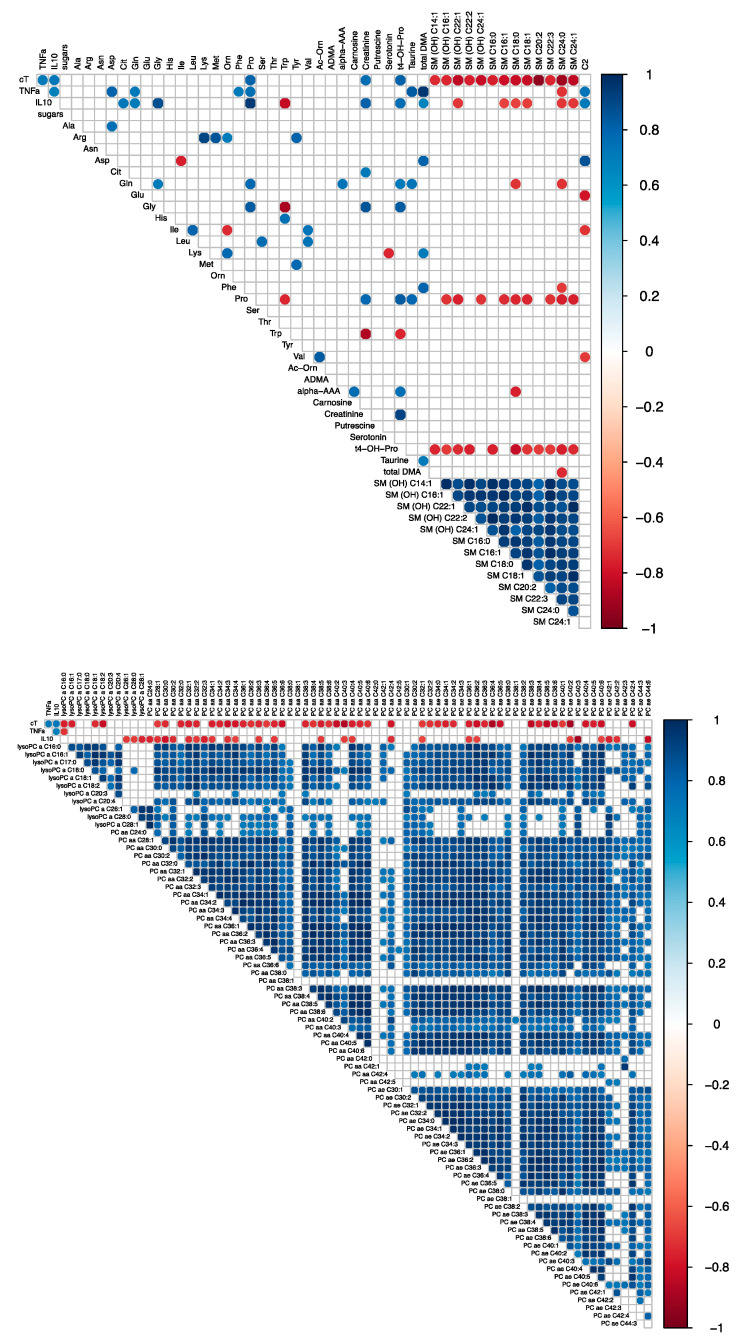
Correlation matrices representing the correlation coefficients between the fold change in concentrations of hs-cTnT, cytokines (TNF-alpha and IL-10) and metabolites evaluated in n = 9 pigs. The metabolites were separated in two matrices for readability purpose. The fold change represents the ratio between the values measured after full resuscitation and baseline (Res./BL). Only the significant correlations are reported (Spearman *p*-value < 0.5). The red dots represent negative correlations, whereas blue dots positive ones. The correlations with cardiac troponin T can be observed in the first row. For the meaning of the abbreviations and metabolites see Appendix A.

**Figure 4 metabolites-12-00319-f004:**
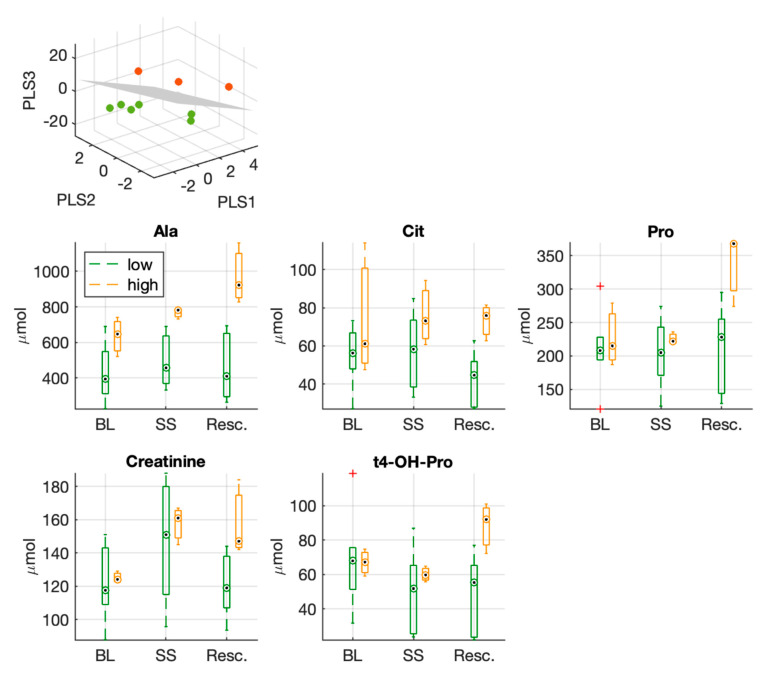
Three-dimensional PLS-DA score plots from the model built with the fold changes (Res./BL) of circulating metabolites used to separate the two experimental groups (low vs. high hs-cTnT group). Below, boxplots of metabolites which resulted significantly different at the end of the experiment (after full resuscitation) between the two groups; proline, creatinine and hydroxyproline were listed among the first ranked features in the PLS-DA VIP score.

**Table 1 metabolites-12-00319-t001:** Clinical and laboratory data at each time point of the animal experiment. Values are reported as median (25th, 75th) percentile for the animals grouped into low and high CT (hs-cTnT) concentration at the end of the experiment. * *p*-value < 0.05 Wilcoxon rank sum test between groups.

	Low hs-cTnT (n = 6)			High hs-cTnT (n = 3)		
	Baseline	Shock	Full Resuscitation	Baseline	Shock	Full Resuscitation
hs-cTnT (ng/L)	16.00 (12.00, 21.00)	21.00 (15.00, 40.00)	26.00 (18.00, 30.00) *	8.00 (8.00, 9.50)	21.00 (18.00, 24.00)	48.00 (42.00, 58.50)
Lactate mmol/L	0.90 (0.90, 0.90)	1.15 (0.90, 1.30)	1.45 (0.90, 1.90) *	0.90 (0.90, 0.90)	2.10 (1.65, 2.55)	3.10 (3.02, 3.55)
HR bpm	74.00 (72.00, 80.00)	94.00 (79.00, 129.00)	148.50 (144.00, 152.00)	82.00 (65.50, 88.75)	160.00 (115.75, 162.25)	150.00 (145.50, 152.25)
DAP rad mmHg	54.50 (51.00, 55.00)	41.50 (37.00, 45.00)	53.50 (48.00, 58.00) *	57.00 (54.75, 58.50)	33.00 (31.50, 36.00)	35.00 (34.25, 40.25)
MAP rad mmHg	71.00 (70.00, 75.00)	54.00 (46.00, 63.00)	76.00 (70.00, 82.00) *	76.00 (73.75, 77.50)	46.00 (45.25, 46.00)	55.00 (51.25, 64.00)
CO mL/min	4600 (4300, 4700)	3200 (2800, 4700)	8550 (7400, 9700)	5400 (4500, 5775)	4850 (3700, 6000)	9900 (9675, 11,100)
SvO_2_ %	62.00 (61.00, 69.00)	55.00 (52.25, 59.25) *	75.50 (73.00, 81.00)	67.00 (61.75, 68.50)	68.00 (65.00, 69.50)	68.00 (64.25, 74.75)
T °C	38.90 (38.70, 39.20)	39.10 (39.00, 39.90)	39.10 (39.00, 39.30)	39.00 (38.85, 39.08)	37.70 (37.70, 38.30)	38.50 (38.28, 39.10)
pH	7.47 (7.46, 7.48)	7.45 (7.41, 7.46)	7.45 (7.45, 7.46)	7.48 (7.47, 7.49)	7.42 (7.41, 7.46)	7.43 (7.35, 7.44)
PaCO_2_ mmHg	47.05 (46.60, 48.00)	48.05 (47.00, 49.50)	47.85 (46.30, 48.40)	45.60 (45.15, 47.48)	48.90 (45.07, 49.95)	48.80 (48.05, 56.15)
PaO_2_ mmHg	122 (114, 130)	122 (110, 143)	130 (101, 165) *	129 (126, 137)	134 (94, 149)	70 (63, 90)
HCO_3_^–^ mmol/L	33.60 (33.00, 34.00)	32.00 (30.70, 33.20)	32.10 (30.00, 33.00)	33.00 (32.25, 34.95)	31.20 (31.20, 31.35)	31.40 (30.20, 31.70)
BE	8.90 (8.80, 9.70)	7.10 (5.20, 8.70)	7.40 (5.20, 8.30)	8.40 (8.10, 10.35)	5.90 (5.82, 6.80)	6.10 (3.85, 6.78)
Sat O_2_ %	99.50 (99.00, 100.00)	99.50 (99.00, 100.00)	99.00 (98.00, 100.00)	100.00 (99.25, 100.00)	100.00 (97.00, 100.00)	89.00 (88.25, 95.75)
Hct %	26.70 (23.00, 30.00)	27.00 (24.00, 33.00)	28.50 (27.00, 30.00)	26.00 (24.50, 27.50)	34.00 (26.50, 38.50)	24.20 (21.05, 30.80)
Na^+^ mmol/L	132 (129, 134)	132 (127, 132)	134 (133, 135)	132 (131, 134)	131 (131, 133)	133 (132, 134)
K^+^ mmol/L	4.40 (4.30, 4.40)	4.30 (4.00, 4.40)	4.35 (4.30, 4.40)	4.00 (4.00, 4.00)	3.80 (3.50, 3.80)	4.30 (4.23, 4.52)
Cl^−^ mmol/L	101 (100, 103)	100 (97, 101)	99 (97, 100)	99.00 (97.50, 102.00)	97.00 (96.25, 98.50)	97.00 (97.00, 98.50)
anion gap	−2.00 (−8.00, −1.00)	−0.35 (−2.20, 5.00)	1.90 (0.00, 6.00)	−1.00 (−1.00, −0.70)	2.80 (1.30, 5.65)	6.20 (2.45, 6.50)
Ca^++^ mmol/L	1.27 (1.24, 1.29)	1.23 (1.18, 1.28)	1.15 (1.10, 1.15)	1.20 (1.15, 1.22)	1.21 (1.13, 1.25)	1.17 (1.03, 1.19)
Glucose mmol/L	90.50 (86.00, 105.00)	88.50 (71.00, 95.00)	93.50 (86.00, 110.00)	94.00 (92.50, 97.75)	62.00 (61.25, 77.75)	64.00 (61.00, 68.50)

## Data Availability

The level of all the measured metabolites in the animal experiment (plasma and heart tissue) organized in a excel table Appendix A. Additional information is available upon request to the corresponding author.

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
