# Peer review of "Metabolites Concentration in Plasma and Heart Tissue in Relation to High Sensitive Cardiac Troponin T Level in Septic Shock Pigs"

_metabolites, 2022, doi:10.3390/metabo12040319_

Round 1
Reviewer 1 Report
Authors investigated metabolomics in septic pigs. They tryed to associate their results with collagen metabolism/degradation, incorporating TnT levels. It is very interesting study, but the followings should be resolved.
Major topic
As the title of this study, “Elevated plasma high sensitive cardiac troponin T in septic shock pigs and collagen metabolism.” Indicates, they should associate the metabolomics results with collagen metabolism/degradation. However, unfortunately, only implication with theories was presented in discussion. As authors commented in limitation section, the elevated level of proline and hydroxyproline in plasma and cardiac tissues could not be associated with collagen metabolism with current results, but they discussed many topics about collagen, even in title.
I think they have cardiac tissues and plasma(maybe), and they could perform at least small study about collagen metabolism/degradation, e.g. histology, biomarker, etc.
Minor topic
- In fig 1, please show the SS group or sham group in figure. e.g. black circle: sham group, blue circle; SS
- Line 213-215. They wrote that there was no altered cardiac performance, but CO indicated that SS w elevated TnT had hyperdynamic cardiac features.
- Fig 4. There are 13 features with high VIP score >1 in Table S5. Why not other variables are compared?
Author Response
Q1: As the title of this study, “Elevated plasma high sensitive cardiac troponin T in septic shock pigs and collagen metabolism.” indicates, they should associate the metabolomics results with collagen metabolism/degradation. However, unfortunately, only implication with theories was presented in discussion. As authors commented in limitation section, the elevated level of proline and hydroxyproline in plasma and cardiac tissues could not be associated with collagen metabolism with current results, but they discussed many topics about collagen, even in title.
A1: We want to thank the reviewer for his/her comments. We agree with reviewer, our results cannot demonstrate a direct association as we clearly stated in the discussion, but they suggest an hypothesis to be explored in further studies.
We reconsidered the title. As the reviewer noticed, it may convey a misleading message about the results of the study, we substituted it with “Metabolites concentration in plasma and heart tissue in relation to high sensitive cardiac troponin T level in septic shock pigs”. We hope the title is now more fitting and representative of the work.
As a matter of fact hydroxyproline and proline play key roles for collagen stability, our study hypothesized an alteration in collagen metabolism, unfortunately we couldn’t prove this association, we rephrased the conclusions in this direction.
Q2:I think they have cardiac tissues and plasma(maybe), and they could perform at least small study about collagen metabolism/degradation, e.g. histology, biomarker, etc.
A2:The experiments were performed in 2017 under the umbrella of the UE project “Shockomics”, unfortunately the frozen tissues have been all utilized and are no more available.
Q3: In fig 1, please show the SS group or sham group in figure. e.g. black circle: sham group, blue circle; SS.
A3: We reviewed figure 1 in order to clearly indicate the values which refer to sham and septic pigs.
Q4: Line 213-215. They wrote that there was no altered cardiac performance, but CO indicated that SS w elevated TnT had hyperdynamic cardiac features.
A4: The hyperdynamic state is a common pathophysiological response in early septic shock related to an inflammatory vasoplegic state, we simply noticed higher HR and CO and more prolonged hypoxia and hypotension in the group with higher concentration of hs-cTnT, but from a clinical point of view, none of the animals had signs of heart failure or of compromised contractility.
Q5:Fig 4. There are 13 features with high VIP score >1 in Table S5. Why not other variables are compared?
A5: The figure 4 refers to the trend of those features which are significantly different between the two groups at the end of the experiment. The PLS-DA model is performed on the fold change between values at end of the experiment and at baseline. The figure wants to highlight that the changes in these metabolites produces also a significant difference between the groups at the end of the experiment.
Reviewer 2 Report
The authors of “Elevated plasma high sensitive cardiac troponin T in septic shock pigs and collagen metabolism.” present an interesting study with the goal of finding correlations between molecular data in different experimental conditions related to the important clinical topic of septic shock. This work is a logical follow up of previous studies from the same authors. A bit of a problem while reading this manuscript is the heavy relegation of methodological details to previously published work from the same authors. It would benefit this manuscript if the authors included explicit, even if brief, recapitulations of some of those details. I consider that that a brief development of the paragraph “Details on the animals, anesthesia, instrumentation and experimental protocol were previously reported [15,16].” (section 2.1) is of particular value. The same can be said about the details about instrumentation in the MS analysis. I recommend that the authors enhance their Methods section with such additional information.
This manuscript is very well written and easy to follow. The findings are very relevant.
There are two major problems with this work that the authors should address:
- I understand that it is difficult to work on high sample sizes, due to animal availability, but considering the stratification applied to the animals, the authors end up with rather unbalanced and low sample size classes (6 vs 3). This leads to problems in all the statistical methods employed by the authors, except the correlations in fig. 3 which are calculated for the whole 9 animals. Is there any other way to stratify the animals to correct this situation?
- The authors should have used the PLS-DA model with greater care: 1- its purpose on this study must be clearly stated. 2- the fixed choice of 3 latent variables must be justified, as, apparently, this number was not optimized in any way. 3- the performance of the model should have been assessed somehow. The authors do not have enough data to perform a good train/test split, but at least a 3-fold CV could have been attempted to show that the PLS-DA model and the derived VIP scores are reliable. The scores plot is useless, in this context. If the PLS-DA model is overfitting, then its use in this paper becomes pointless.
Other minor issues:
- In the legend of figure 1, the explanation of the colors of the points should be provided. Furthermore, the lines represent exact levels and, to be mathematically rigorous, the legend should read, for example “lactate = 2 mmol/L.” instead of “lactate >2 mmol/L.”
- Figure 2 is quite complex with 24 panels. As a consequence, the box-plots become too thin and hard to read, especially when outliers are indicated. Could the authors make the boxes a little wider(they are thinner than the diameter of the data points)?
Author Response
Q1:The authors of “Elevated plasma high sensitive cardiac troponin T in septic shock pigs and collagen metabolism.” present an interesting study with the goal of finding correlations between molecular data in different experimental conditions related to the important clinical topic of septic shock. This work is a logical follow up of previous studies from the same authors.
A1: We thank the reviewer for his/her appreciation of our study.
Q2: A bit of a problem while reading this manuscript is the heavy relegation of methodological details to previously published work from the same authors. It would benefit this manuscript if the authors included explicit, even if brief, recapitulations of some of those details. I consider that that a brief development of the paragraph “Details on the animals, anesthesia, instrumentation and experimental protocol were previously reported [15,16].” (section 2.1) is of particular value. The same can be said about the details about instrumentation in the MS analysis. I recommend that the authors enhance their Methods section with such additional information.
A2: We added additional information about the experiments in the Methods section. We tried to summarize the key points as suggested by reviewer.
Q3: This manuscript is very well written and easy to follow. The findings are very relevant.
There are two major problems with this work that the authors should address:
I understand that it is difficult to work on high sample sizes, due to animal availability, but considering the stratification applied to the animals, the authors end up with rather unbalanced and low sample size classes (6 vs 3). This leads to problems in all the statistical methods employed by the authors, except the correlations in fig. 3 which are calculated for the whole 9 animals. Is there any other way to stratify the animals to correct this situation?
A3: We are aware about the small sample size limitation, for the univariate statistical analyses we provided figures which display the values for each pig and not only summary overview like boxplot. For the multivariate analyses it’s unreliable to use any method for dealing unbalanced classes, which are not the simple over-sampling or under-sampling. The PLS-DA was adopted only to see if the groups are separable on the basis of metabolites concentration, the purpose was not to develop a classification model.
Q4: The authors should have used the PLS-DA model with greater care: 1- its purpose on this study must be clearly stated. 2- the fixed choice of 3 latent variables must be justified, as, apparently, this number was not optimized in any way. 3- the performance of the model should have been assessed somehow. The authors do not have enough data to perform a good train/test split, but at least a 3-fold CV could have been attempted to show that the PLS-DA model and the derived VIP scores are reliable. The scores plot is useless, in this context. If the PLS-DA model is overfitting, then its use in this paper becomes pointless.
A4: The use of 3CV means to have folds with 1 pig of the positive class and 2 of negative class, it would be impossible to get a representative model. The PLS-DA was not meant to develop a classification model, we’re aware that the model is overfitting our data, but the purpose was to see if the metabolites concentration were able to separate the group and to see which features played a major role in the model so to infer possible associations. Figure 4 shows the univariate analyses on some of that features which confirm a different trend in the groups.
Q5: In the legend of figure 1, the explanation of the colors of the points should be provided. Furthermore, the lines represent exact levels and, to be mathematically rigorous, the legend should read, for example “lactate = 2 mmol/L.” instead of “lactate >2 mmol/L.”
A5: we modified figure1 according to the suggestions of the reviewer#1 and we revised the legend as well.
Q6:Figure 2 is quite complex with 24 panels. As a consequence, the box-plots become too thin and hard to read, especially when outliers are indicated. Could the authors make the boxes a little wider(they are thinner than the diameter of the data points)?
A6: Figure2 was substituted with a new one, the boxplot are wider as before, we hope now it is easier to look at value distribution.
Round 2
Reviewer 1 Report
Authors revised the manuscript as they could do.
Reviewer 2 Report
The authors have improved their manuscript and have addressed all the issues in their response.
Although the manuscript does not mention it explicitly, I disagree with the justifications provided about the use of the PLS-DA in their response. The authors acknowledge that the model might be overfitting the data and state that "The PLS-DA was adopted only to see if the groups are separable on the basis of metabolites concentration, the purpose was not to develop a classification model" and, later, "The PLS-DA was not meant to develop a classification model, we’re aware that the model is overfitting our data, but the purpose was to see if the metabolites concentration were able to separate the group and to see which features played a major role in the model so to infer possible associations"
Hoping that I am not quoting the authors out-of-context, I must reinforce the idea that PLS-DA scores plots do not demonstrate global metabolic differences between classes, samples, or groups, although it looks like it does. PLS-DA scores plots almost always exhibit separation between classes, because, using the encoding of class information, not significant variables can have a large weight in the computation of the latent variables, in a way that suggests metabolic (biological) differences. We can even scramble class membership with nonsensical labels using the same data and obtain an apparent class separation in scores plots. Unsupervised methods (PCA) do not use class membership and can much more reliably indicate group metabolic separations.
As long as the authors do not introduce the interpretation stated in their response to this reviewer in the manuscript, then nothing is really wrong, although the use of the PLS-DA model in their work turns out to be almost useless. The first panel of figure 4 does not show what the authors have stated in their response.
I thank the authors for their great effort in making summaries of the details of the procedures and for the improvement of the box-plots.